# Effect of Temporary Cement, Surface Pretreatment and Tooth Area on the Bond Strength of Adhesively Cemented Ceramic Overlays—An In Vitro Study

**DOI:** 10.3390/dj11010019

**Published:** 2023-01-05

**Authors:** Sanita Grinberga, Evaggelia Papia, Jolanta Aleksejuniene, Vita Zalite, Janis Locs, Una Soboleva

**Affiliations:** 1Department of Prosthetic Dentistry, Faculty of Dentistry, Riga Stradins University, LV-1007 Riga, Latvia; 2Department of Materials Science and Technology, Faculty of Odontology, Malmö University, 205 06 Malmö, Sweden; 3Faculty of Dentistry, University of British Columbia, 2199 Wesbrook Mall, Vancouver, BC V6T 1Z3, Canada; 4Rudolfs Cimdins Riga Biomaterials Innovations and Development Centre, Faculty of Materials Science and Applied Chemistry, Institute of General Chemical Engineering, Riga Technical University, LV-1007 Riga, Latvia

**Keywords:** adhesive cementation, dental overlay, dental on-lay, enamel, surface pre-treatment, temporary cementation, zirconia, cement, dentistry

## Abstract

Several viewpoints have been reported regarding the effect of temporary cements, different surface pretreatment protocols before adhesive cementation, and predictive factors. This in vitro study tested if temporary cement, pretreatment of the tooth surface, the size of enamel or dentine influence adhesive cementation to zirconia ceramics. Twenty premolars were prepared for determination of enamel and dentin area, bond strength test and failure analysis. The samples were divided into two groups: untreated prior adhesive cementation (n = 10) and with temporary cementation done, pretreated prior adhesive cementation (n = 10). Zirconia overlays *(Katana Zirconia STML)* were cemented on the grounded flat teeth surfaces using Panavia V5. An additional six premolars underwent dentine tubule analysis with SEM to detect temporary cement residues after temporary cementation on an untreated tooth surface (n = 3) and on a pretreated surface (n = 3). The independent sample t-test was used to compare the two groups and the means of the total tooth, dentin or enamel areas did not differ significantly between the untreated and pretreated specimens. The mean tensile bond strength was significantly (*p* = 0.005) higher in the pretreated specimens (337N) than in the untreated ones (204N). The overall multivariable linear regression model with three predictors (surface pre-treatment, enamel area and dentine area) was significant (*p* = 0.003), among which the size of enamel was the strongest predictor (β = 0.506; *p* = 0.049), followed by the pretreatment effect (β = 0.478; *p* = 0.001) and the size of dentin area (β = −0.105; *p* = 0.022).

## 1. Introduction

Over recent years adhesive cementation has been increasingly used, particularly for aesthetic dental restorations, mainly due to improved adhesion, optical properties common to contemporary ceramic materials and the growing demand for cosmetic restorations [1]. The use of resin luting cements for indirect ceramic restorations has increased because they can bond to tooth and restoration and are less soluble. Consequently, their retention is better than that of non-adhesive luting cements [2]. In addition, as compared with direct composite restorations, indirect bonded ceramic restorations can better restore proximal contacts, replicate occlusal morphology, ensure marginal accuracy and lead to less cement contraction [3]. However, ensuring adequate bonding is generally complex, especially when restorations must be inserted posteriorly [4]. Thus, in certain clinical situations, it is recommended to first restore with temporary restorations. Furthermore, most ceramic restorations require extensive dental laboratory work, thus until the final cementation temporary cements are used to minimize tooth sensitivity and the risk of infection or tooth migration [5].

Different cement types are used to ensure the retention of temporary restorations, commonly chosen for their soothing effect, easy handling and relatively low cost [6]. However, temporary restorations could adversely impact the bonding strength of permanent restorations [7]. The detrimental impact of the temporary cements containing eugenol on adhesive systems was reported, attributed to eugenol being a radical scavenger inhibiting the polymerization of resin-containing cements [8]. Regarding the bonding to tooth surface after the use of zinc oxide eugenol-containing cements, some studies found no reduction of overall bond strength [5], while other studies reported a reduction of the bond [3]. Concomitantly, it is important to consider that even eugenol-free cement may reduce the bonding to dentine [5]. Furthermore, it is important to consider that reduction in bonding is not necessarily due to eugenol but may be caused by other residuals such as mineral oil particles or other ingredients [9]. In addition, to weaken the effects of temporary cements, some dentists use petroleum jelly, which is also recommended by some manufacturers. To sum up, more evidence is needed to better understand the impact of permanently cemented prosthetic restorations.

It should also be taken into consideration that teeth have two different hard tissues, enamel and dentin, each having different properties for cementation [10]. The amount and quality of enamel have a critical role when it comes to adhesive cementation and the literature shows that restorations can be luted even on a flat tooth surface, as long as the size of the enamel ring is sufficient [11]. This applies to modern minimally invasive preparation techniques when teeth are not prepared into a macro-mechanical retention shape, and where bonding mainly relies on chemical retention [6]. To restore the anatomy and interface of the tooth, overlays, on-lays and endo-crowns are routinely used. Until such restorations are cemented permanently, teeth are closed with temporary materials, but this may impact the quality of the permanent cementation.

The current experimental study tested if the use of temporary cement, pretreatment of the tooth surface and the size of enamel or dentin influence the quality of adhesive cementation. The first study hypothesis (quantitative) was that surface pretreatment and differences in enamel or dentin areas impact the bonding between the zirconia restoration and tooth. The second study hypothesis (qualitative) was that due to the use of temporary cement residual cement particles will always be present in the dentine tubules.

## 2. Materials and Methods

### 2.1. Inclusion of Samples

The study was approved by the Institutional Research Ethics Board (No: 6-3/5/46). The sample comprised premolars which were extracted due to orthodontic or periodontal indications during the period from September 2020 to February 2021. Only a total of 26 intact premolars without caries, extensive restorations, abrasions, cracks and hypocalcifications were collected. The ultrasonic scaler (PiezoLED, KaVo Dental, manufactured by E.M.S. Electro Medical Systems S.A., Nyon, Switzerland) was used to clean these teeth from plaque, calculus and periodontal tissues. After their cleaning, teeth were put into the physiologic saline solution and stored in a refrigerator at a temperature of 4°C and tested in less than 6 months after their extraction [3,5]. The experimental setting of the study is visualized in Figure 1.

### 2.2. Specimen Preparation for the Surface Analysis and Bond Strength Test

We chose the number of specimens needed for our study based on the setting-based information previously provided in a similar experimental study where the tensile bond strength test and surface analysis were used [3,5]. Twenty pre-selected teeth (n = 20) were embedded in a chemically cured acrylic resin (PMMA SR Ivocap high impact, Ivoclar Vivadent) 16 mm × 25 mm cylinder, strictly in the center and parallel to the tooth’s vertical axis. To ensure adequate retention, depth grooves were made on the root surfaces and the teeth were grounded flat horizontally (Edenta Superflex diamond disk, Fine, Edenta) under water cooling to expose dentine surfaces approximately 1.5 mm from the pulp (Figure 2a–c).

### 2.3. Measuring Enamel and Dentin Areas

For tooth area measurements, all specimens were placed on a millimeter paper at the same distance (20 cm) and then photographed under standardized conditions (Canon EOS 80D 60 mm f/10, 1/40 s, ISO 400, flash compulsory). The enamel and dentine areas were calculated using the Image Pro software (2D Image Capture Module, Media Cybernetics Inc., Rockvill, MD, USA) (Figure 3). First, for each specimen, the contours of the enamel and dentine areas were marked manually to enable further evaluation by the software; for each specimen the sizes of the enamel and dentine areas (mm^2^) were calculated.

### 2.4. Manufacturing of Overlay

The Ceramill Mind (Amann Girrbach AG, Koblach, Austria) was used to design 20 zirconia cylindrical shape overlays (Katana Zirconia STML, Kuraray Noritake, Tokyo, Japan), the diameter of each was 13.2 mm and the height was 3.2 mm, also considering the potential shrinkage during the sintering [12]. Further procedures were caried out following the manufacturer’s protocol for crystallization and firing [13].

### 2.5. Division of Specimens in Preparation for Testing the Bond Strength

Before cementation, teeth specimens (n = 20) were allocated to one of the two groups (Figure 1). The pre-treated specimens (n = 10) received temporary acrylic overlays (Success CD, Promedica Dental Material GmbH, Neumunster, Germany) which were cemented with Temp Bond NE (Kerr Dental Kerr Corporation, Orange, CA, USA) cement using 15N pressure for 60 seconds to ensure an equally calibrated force for all specimens (Figure 4a,b). After the temporary cementation, these specimens were kept for one day at room temperature in a plastic container with a moistened napkin. The following day, the acrylic overlays were removed, and the residual temporary cement was scraped off with a probe and polished with pumice (Detartine pumice, Septodont, St. Maur des Fosses, France, RDA 143.63 ± 23.86) and a rotating brush, then rinsed with water until visual inspection confirmed that there was no cement left on the surface. This experimental setting was meant to represent the clinical situation, where temporary cement is used and the tooth surface is pretreated prior to the subsequent adhesive cementation. All procedures were performed by one operator. 

The second group (untreated specimens, n = 10) prior to adhesive cementation did not receive any pretreatment.

### 2.6. Adhesive Cementation

Prior to cementation, the zirconia overlays were sandblasted (AlO_3_ 50 μm at a pressure of 2 Ba) for 15 s at a distance of 10 mm [14,15] and subsequently cemented with the Panavia V5 (Kuraray Noritake, Tokyo, Japan) system following the manufacturer’s instructions [16]. One operator cemented all specimens (n = 20) applying the finger pressure, subsequently scrapping the cement surplus with an applicator, while checking visually that the residual cement is fully removed, prior to its subsequent polymerization (Figure 4c). Then, the specimens were kept in a humid environment for a week [17].

### 2.7. Tensile Bond Strength Test

The tensile bond strength test measured the bond strength, employing the Instron Universal Machine (Instron® Mechanical Testing Systems, Norwood, MA, USA) where the head speed was set at 0.5 mm/min (Figure 5a–c) [18]. Each specimen was mounted in the testing machine using a custom-made holder which was centered and levelled to ensure that the force was evenly applied. The load at failure was indicated in Newtons (N). The test was performed blindly meaning that the operator did not know to which study group any of the specimens belonged.

### 2.8. Failure Mode Analysis

When the tensile bond strength test was completed, the specimens, teeth and overlay surfaces, were qualitatively assessed under 10 × magnification with a microscope (Wild M3, M7A Wild Heerbrugg, Heerbrugg, Switzerland). Failure modes differed among the specimens; thus, their mode of fracture was categorized either as adhesive type or cohesive type. An adhesive fracture was defined when the remnant of the cement was present either on the overlay or on the tooth surface. We considered a cohesive fracture when the failure was within the same material, i.e., either the tooth fractured, or the cement was present on both the overlay and the tooth surfaces [19].

### 2.9. Specimen Preparation for Dentine Tubules Substance Analysis

The remaining six teeth were fixed in acrylate blocks (Pattern Resin LS, GC America Inc., Alsip, IL, USA) and grounded (Superflex Diamond disk, Fine, Edenta) flat under water-cooling to expose the dentine surfaces. To ensure equally calibrated force for all specimens, temporary acrylic overlays (Success CD, Promedica Dental Material GmbH, Neumunster, Germany) were cemented with a Temp Bond NE (Kerr Dental Kerr Corporation, Orange, CA, USA) using 15N pressure for 60 s. Then, the specimens were kept in a plastic container with a moistened napkin for one day at room temperature. Subsequently, the temporary overlays were detached from three specimens, the cement residues from which were removed with a dental instrument and subsequently these specimens were polished by pumice with a rotating brush until visual examination confirmed no cement residuals. For the other three samples, cement residues were intentionally left untouched. Then, all six specimens were rinsed with water and mechanically broken along the vertical axis prior to their scanning electron microscopy (Tescan Mira/LMU, Brno-Kohoutovice, Czech Republic) with energy-dispersive X-ray (EDX) analysis (Oxford Instruments X-MaxN detector size 150 mm2, Oxford, United Kingdom) to detect Zn ions known to be present in temporary cements [20]. The latter information was available from the safety sheet provided by the manufacturer. Zn ions are only present as the inorganic substance in the form of component zinc oxide (CAS No. 1314-13-2; an amount ranging 60–100%). Thus, finding Zn inorganic particles in the cross section of the dentine tubules indicated the residues of temporary cement.

### 2.10. Statistical Analyses

All data were analyzed employing the SPSS version 27.0 software (IBM, New York, NY, USA) and the threshold for significance for all tests was set to *p* < 0.050. The Kolmogorov-Smirnov tested the normality assumption in preparation for the subsequent inferential analyses. Independent sample t test compared the mean bond strength values between the two experimental groups. The distribution of different types of failure modes between the two groups was analyzed with a two-sided Fisher Exact test. The multivariate linear regression analysis tested three potential predictors of bond strength: (1) enamel area, (2) dentin area, and (3) the effect of tooth surface pretreatment prior to adhesive cementation. 

## 3. Results

### 3.1. Influence of Tooth Area on the Bond Strength

The means of the total tooth area, dentine and enamel areas are compared between the two experimental groups in Table 1, where one can see that any of these means were not significantly different between the untreated and treated groups of specimens. The mean tensile bond strength was significantly (*p* = 0.005) higher in the pretreated group than the corresponding mean in the untreated group.

The multivariable linear regression model tested three predictors: surface pretreatment, the size of the enamel and dentine areas and their joint effect to explain variations in the bond strength (Table 2). The overall prediction model was significant (*p* = 0.003), and the three predictors jointly explained 57.7% (Adjusted R square = 0.577) of variation in the bond strength at the time of the specimen fracture. All three predictors were significant, among which the size of enamel was the strongest predictor (β = 0.506; *p* = 0.049), followed by the pretreatment effect (β = 0.478; *p* = 0.001) and the size of dentin area (β = −0.105; *p* = 0.022).

### 3.2. Failure Modes 

During the tensile bond strength test, the specimens failed differently, and there were significant (The Fisher´s Exact test, *p* = 0.011) proportional differences between the two experimental groups in the types of failures, where the untreated specimens experienced only adhesive failures (100%), while the pretreated specimens experienced both type of failures, of which 40% were adhesive and 60% were cohesive (Figure 6a–c). 

### 3.3. Qualitative Substance Analysis of Cement Residues Present in Dentine Tubules

Zn ions were detected in all three untreated specimens (100%), while only one of the three pretreated specimens (33%) showed the presence of such ions (Figure 7a,b).

## 4. Discussion

In most cases, indirect aesthetic restorations prior to their final cementation require the restorations to be fixed with a temporary cement, to ensure that the prepared tooth surfaces are protected from bacterial invasion and tooth sensitivity and the tooth function is maintained until the laboratory and clinical procedures are completed and the final restoration can be cemented [5]. To limit the penetration of temporary cement into the dentinal tubules immediate dentin sealing can be used. However, using dentine sealing would not allow us to answer the question of our study. Therefore, we evaluated how the use of temporary materials, surface pre-treatment and the size of the total tooth area (enamel and dentin) available for cementation may impact the bonding between the tooth and the ceramic restoration.

Both study hypotheses were confirmed; the first hypothesis stated that the surface pre-treatment and differences in enamel or dentin areas all independently impact the bonding; and the second hypothesis stated that cement residues are left in dentine tubules when temporary cement is used.

Our experiment showed that the bond strength was higher in the pre-treated group, in which the temporary cement was mechanically removed and subsequently the tooth surface polished with the pumice. This finding indicates that cleaning temporary cement residues is important: concomitantly one also needs to consider that having more enamel available for cementation has an even stronger impact than the pre-treatment on the bond strength of adhesive cementation.

Contradictory evidence is available regarding the effect of the temporary cement residues, their removal methods and their potential impact on bond strength, where some studies found that the cleaning method of temporary cement contributes to the strength of the adhesive bond [21,22], while other studies did not find such the impact [5,6]. Different protocols for cleaning tooth surfaces after the use of temporary restorations were suggested, for example, manual removal of cement residues using an excavator and/or mechanical removal of cement by polishing with pumice using a rotational brush. Other suggested methods were sandblasting with aluminium oxide particles and chemical cleaning with cleaning agents such as hydrogen peroxide or chlorhexidine [5,6,21,23,24]. Elimination of remnants of temporary cement with or without eugenol by excavator led to a higher bonding strength as compared to sandblasting [5]. Furthermore, Watanabe et al. reported decreased bond strength due to temporary cement, but increased bond when dentine conditioners were used [25]. In the current study, both manual and mechanical cleaning were chosen as per our clinic’s protocol; seemingly our study findings support that this is an important step towards increasing the bond strength. Considering the time factor of when the pre-treatment of the cementation surface takes place, Al-Akhali et al. reported that a newly cleaned zirconia surface has a higher bond strength [15].

Of importance, both the SEM analysis and optical microscopy showed that cement residuals remained on dentin surfaces even after their mechanical cleaning combined with etching with 37% phosphoric acid [25]. In the current study, the examination of the dentin surface by SEM analysis confirmed that, even when the temporary cement was mechanically removed, cleaned and polished from the tooth surface, zinc ions were still found in the dentin tubules. This finding may indicate that the residual particles from temporary cement always penetrate the dentine tubules. Interestingly, Peutzfeldt and Asmussen reported that residues from both eugenol-containing or non-eugenol cements showed no influence on the bonding strength between adhesive restorations and dentine [26]. Similar findings were reported in other studies which tested several other adhesive systems such as Syntac (*Ivoclar Vivadent*), Scotchbond Multipurpose Plus (*3M ESPE*), Panavia F 2.0 (*Kuraray Noritake*), Exite/Variolink II (*Ivoclar Vivadent*) [5]. 

The present study did not observe the impact of the residual cement on the bond strength of the adhesive cementation. However, one has to practice caution when comparing the findings across the studies, because in some studies the cementation area included both the dentine and enamel tissues, while in other studies, only dentin or other substrates for adhesive cementation were used. In our study, we evaluated the importance of the enamel area and demonstrated that having more enamel contributes to higher bond strength. Given the different results in the literature, seemingly no consensus has yet been reached.

It is known that the way the specimens are stored or how they are sterilized may influence the outcomes of in vitro studies conducted on extracted teeth. For example, the usage of saline and 5.25% NaClO significantly lowered the shear bond strength between the composite and dentin, consequently such storage media were not recommended. Furthermore, sterilization as well as autoclaving of teeth primarily kept in saline and formalin solutions also had negative effects as compared to specimens stored only in formalin [27]. Following the advice that specimens need to be hydrated throughout the experiment, we stored our samples in saline solution during teeth collection and a humid environment was also maintained during specimen preparation and their cementation. In addition, we followed another recommendation for in vitro testing, namely, not to extend the storage time of extracted teeth beyond 6 months [28]. The main factor influencing the result was the size of the enamel ring in the tooth substrate. 

A limitation of our study is the small sample size, due to our limited availability of biological material. For standardization purposes, we needed to choose only intact human premolars with complete root formation to test the tensile bond strength. In addition, to ensure equivalent test conditions for all samples, an original fixation method was created. Since the anatomy is specific to individual root parts, slight deviations might have occurred in positioning the longitudinal axis of the teeth parallel to the vertical axis of the cylinders. To make this process as uniform as possible, this was done by one operator. 

Since our interest was focused on studying the effect of the provisional cementation, it was important to apply equal force to create an equivalent impact on the dentinal tubules employing a custom-made press. Unfortunately, the same procedure could not be applied for adhesive cementation, which is more technic-sensitive, and consequently requires quick actions. In the current study, overlays were adapted on a tooth surface with an instrument and using finger pressure. Ultrasound was successfully used in previous in vitro studies in reducing the gap between the tooth and ceramic [29]. To ensure proper seating, we used the following recommendations: pressure 1.25 MPa for 3 minutes during adhesive cementation, and the tensile strength test used a device with the load applied perpendicular to the specimen. This was an important consideration as unequal stress distribution can create shear and bending forces on the overlay and tooth, including their interface. This may explain the cohesive fracture of one tooth we observed [30].

The Panavia V5 (*Kurary Noritake*) cement system containing 10-methacryloyloxydecyl-dihydrogen-phosphate (10MDP), was chosen as the permanent cement. This decision was based on the evidence that a 10MDP-based system showed a higher bond strength than other cement systems [31]. Furthermore, the 10MDP-based system provides stable bonding between different materials, such as teeth, metal, methacrylate base restorations and ceramic as zirconia, as well as displaying low performance for sensitivity [32]. In terms of water sorption, the Panavia V5 also showed good results [33]. Due to the known excellent cementation performance between zirconia and the tooth surface, zirconia was chosen as the basis because of its good mechanical properties as well as its aesthetic properties that favour its use in restoring posterior teeth.

Nevertheless, another limitation of this study is the lack of cyclic fatigue that could outline even more the behaviour of the adhesive cementation interface and the external gap progression [34]. Additionally, other variables could have an influence on bonding such as enamel pre-treatment [35] or sandblasting [36]. Future research should take such factors into consideration while designing their experiments.

To measure the adhesion surface area of the samples, a photography method was used and data analysed with the Image Pro software. Since this operation was performed only once, a small deviation was possible; however, the postponement of manual points was done in increments. A small error occurred, but this should not have a substantial influence on the results, as all data recordings were made by the same, calibrated operator who was able to accurately differentiate between enamel and dentin areas.

Although several previous studies focused on cleaning methods and the effect of eugenol on bonding, there is still no consensus concerning clear guidelines for dentists to support their clinical decisions [7,26,37].

## 5. Conclusions

Within the limitations of this in vitro study, we conclude that the bonding strength can be determined by the size of enamel, dentine and surface pre-treatment prior to adhesive cementation.

## Figures and Tables

**Figure 1 dentistry-11-00019-f001:**
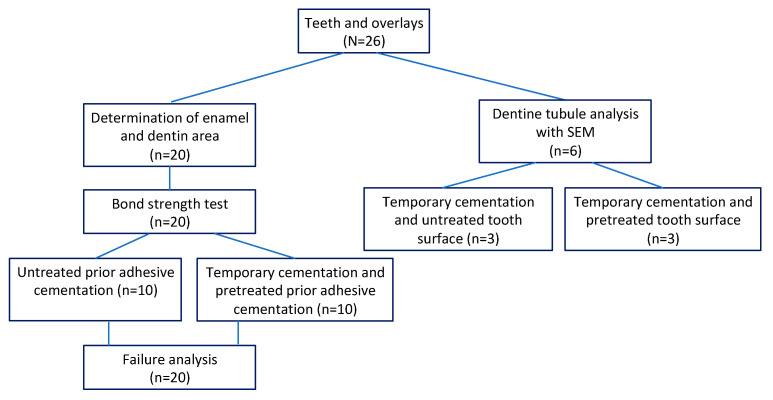
A flow chart of the study experimental design.

**Figure 2 dentistry-11-00019-f002:**
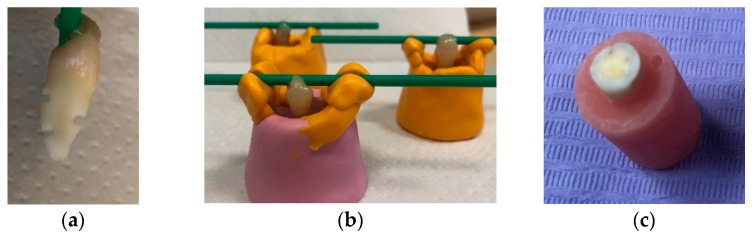
Tooth preparation steps: (**a**) Retention grooves on the root surface; (**b**) Specimens during curing; (**c**) Specimens after preparation.

**Figure 3 dentistry-11-00019-f003:**
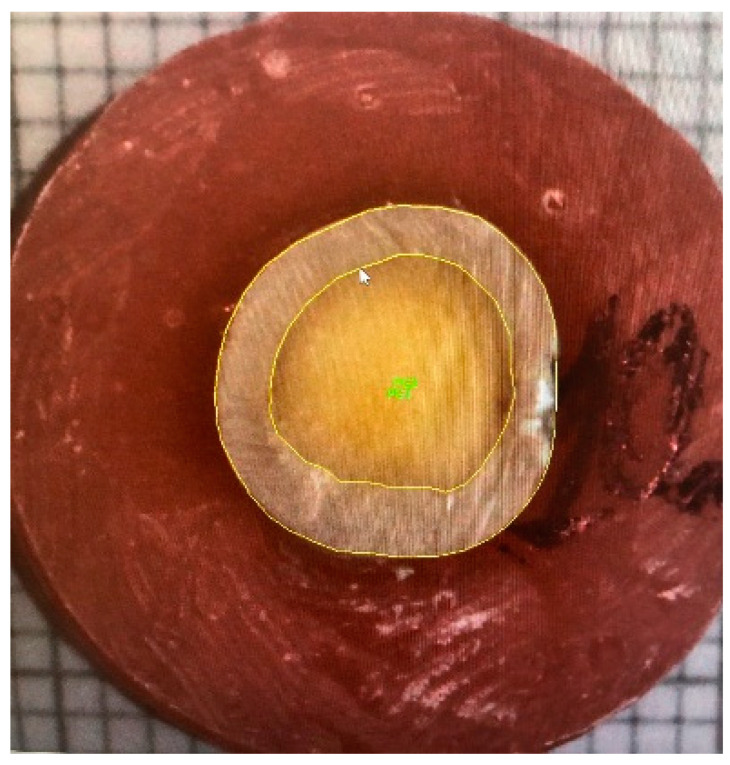
Measuring enamel and dentin areas. The area of enamel and dentine including the border line is marked respectively.

**Figure 4 dentistry-11-00019-f004:**
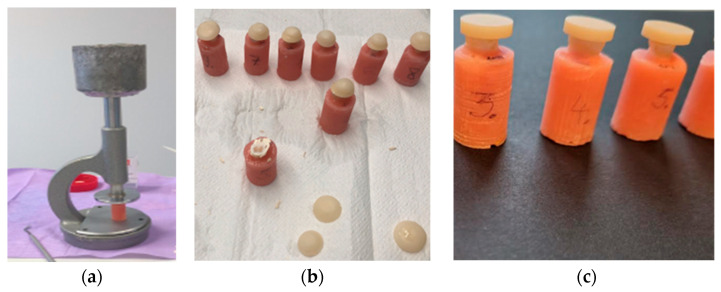
(**a**) Customized press (15N) for temporary overlay cementation. (**b**) Process of temporary cementation. Removal of temporary overlays and cement residues. (**c**) Specimens after adhesive cementation.

**Figure 5 dentistry-11-00019-f005:**
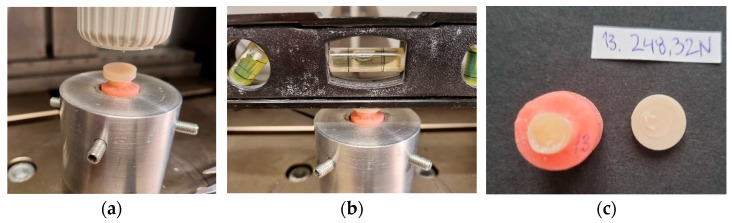
Tensile bond strength test. (**a**) Specimen adapted in a holder; (**b**) The specimen levelled and passed selection; (**c**) Specimen after the test and for failure mode analysis.

**Figure 6 dentistry-11-00019-f006:**
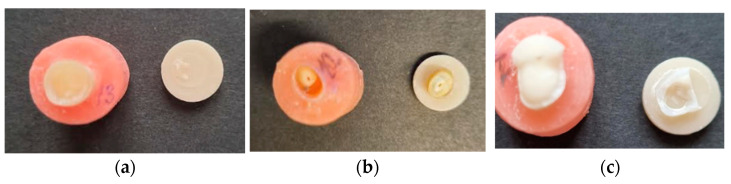
Different failure modes. (**a**) Adhesive fracture, overlay separated from the tooth evenly with cement on the overlay, without any visible fractures; (**b**) Cohesive fracture: the failure includes a total specimen fracture or (**c**) cohesive with cement present on both surfaces, overlay and tooth specimen.

**Figure 7 dentistry-11-00019-f007:**
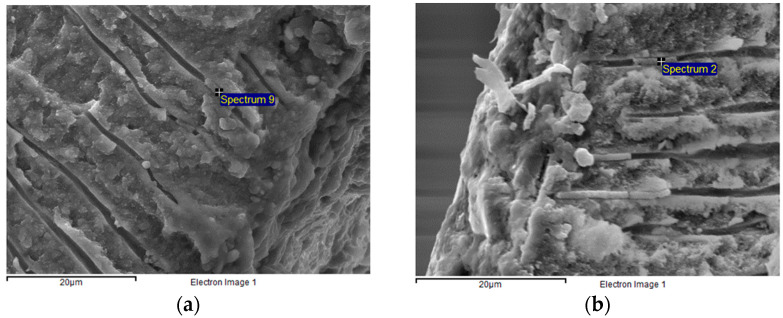
Zn ions detected in the dentine tubules (SEM analysis). (**a**) Untreated specimen; (**b**) Pretreated specimen.

**Table 1 dentistry-11-00019-t001:** The total tooth area, enamel and dentin area and bond strength–comparisons between the untreated group (n = 10) and pretreated group (n = 10) specimens.

	Untreated Group	Pretreated Group	*p* Values #
	Mean (SD)	Mean (SD)	
Total tooth area (mm^2^)	59.4 (15.0)	60.1 (10.6)	0.913
Dentin area (mm^2^)	33.4 (7.9)	30.5 (4.6)	0.319
Enamel area (mm^2^)	26.0 (9.2)	29.6 (7.4)	0.347
Bond strength (N)	203.6 (59.2)	337.4 (116.7)	0.005

# Independent sample *t*-test.

**Table 2 dentistry-11-00019-t002:** Predictors of bond strength (Multivariable Linear Regression Model).

Outcome: Bond Strength (N) at Time of Specimen Fracture Model Summary: *p =* 0.003, Adj. R^2^ = 0.577.				
PREDICTORS	Unstand. Coef (95% CI)	Stand. Coeff. β	*p* Values	Tolerance
Untreated vs. treated	106.8 (21.1; 192.6)	0.478	0.001	0.807
Dentin area (mm^2^)	−1.9 (−9.4; 5.6)	−0.105	0.022	0.669
Enamel area (mm^2^)	7.0 (1.2; 12.8)	0.506	0.049	0.673

## Data Availability

The data that support the findings of this study are available from the corresponding author, S. Grinberga, upon request.

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
