# Peer review of "Effect of Temporary Cement, Surface Pretreatment and Tooth Area on the Bond Strength of Adhesively Cemented Ceramic Overlays—An In Vitro Study"

_dentistry, 2023, doi:10.3390/dj11010019_

Round 1
Reviewer 1 Report
Dear Authors,
the paper is about a very interesting topic.
Here are some suggestions to improve the manuscript.
Lines: 47-9
The authors wrote:
“Thus, in certain clinical situations, it is recommended to first restore with temporary restorations, so that both patients and dentists may assess the restoration appearance and function over a longer period of time.”
The reviewer thinks that “, so that both patients and dentists may assess the restoration appearance and function over a longer period of time.” can be safely removed.
Line 62:
use: “to the use of eugenol”
Line 82:
Please rephrase the following sentence, it’s not clear:
“The first study hypothesis (quantitative) stated that surface pretreatment, differences in enamel or dentin areas impact the bond strength of cementation between the zirconia restoration and tooth.”
Figure 1:
there are some formatting issues: a question mark appears in every box.
Materials and methods:
It is not very clear what the following sentence means:
The second group (untreated specimens, n=10), prior to adhesive cementation, was not subjected to the above described manipulations.
Does it mean that the teeth were prepared and overlays were cemented afterwards?
Please specify better to the reader what was or was not performed on the specimens
Another limitation to be added is relative to the lack of cyclic fatigue analysys.
The sentence that could be added could be something like: “Nevertheless, another limitation of thi study is the lack of cyclic fatigue that could outline even more the behavior of the adhesive cementation interface and the external gap progression”
The authors could cite the following paper that support the above mentioned sentence: [https://doi.org/10.1111/jerd.12837 ]
The authors have not treated the specimen with IDS (Immediate dentin sealing) which is the gold-standard for indirect adhesive cementation.
This could also have avoided the penetration inside the tubules.
Please support the decision not to use IDS.
Author Response
Our responses to Reviewer 1
We appreciate your helpful comments and suggestions which allowed us to improve the manuscript. We attentively reviewed all suggestions and revised our manuscript accordingly. Below we present our point-by-point responses, in addition to changes highlighted in the revised manuscript. Thank you for your time and expertise.
Reviewer comment #1:
Lines: 47-9 The authors wrote: “Thus, in certain clinical situations, it is recommended to first restore with temporary restorations, so that both patients and dentists may assess the restoration appearance and function over a longer period of time.” The reviewer thinks that “, so that both patients and dentists may assess the restoration appearance and function over a longer period of time.” can be safely removed.
Our response: Thank you for the advice, we revised the manuscript as follows “Thus, in certain clinical situations, it is recommended to first restore with temporary restorations.” (Line 50-52)
Reviewer comment #2:
Line 62: use: “to the use of eugenol”
Our response: We revised according to your suggestion. “Furthermore, it is important to consider that reduction in bonding is not necessarily due to use of eugenol, but may be caused by other residuals such as mineral oil particles or other ingredients” (Line 71-72).
Reviewer comment #3:
Line 82: Please rephrase the following sentence, it’s not clear:
“The first study hypothesis (quantitative) stated that surface pretreatment, differences in enamel or dentin areas impact the bond strength of cementation between the zirconia restoration and tooth.”
Our response: We clarified information about both hypotheses as follows:
“The first study hypothesis (quantitative) was that surface pretreatment, differences in enamel or dentin areas impact the bonding between the zirconia restoration and tooth. The second study hypothesis (qualitative) was that due to the use of temporary cement the residual cement particles will always be present in the dentine tubules.” (Line 90-96)
Reviewer comment #4:
Figure 1: there are some formatting issues: a question mark appears in every box.
Our response: We submitted Figure 1 in tiff format.
Reviewer comment #5:
Materials and methods:
It is not very clear what the following sentence means:
The second group (untreated specimens, n=10), prior to adhesive cementation, was not subjected to the above described manipulations. Does it mean that the teeth were prepared and overlays were cemented afterwards? Please specify better to the reader what was or was not performed on the specimens.
Our response: We revised accordingly, clarifying the information about the treatment. “The second group (untreated specimens, n=10), did not receive any pretreatment prior to adhesive cementation.” (Line 162)
Reviewer comment #6:
Another limitation to be added is relative to the lack of cyclic fatigue analysis. The sentence that could be added could be something like: “Nevertheless, another limitation of this study is the lack of cyclic fatigue that could outline even more the behavior of the adhesive cementation interface and the external gap progression”
The authors could cite the following paper that support the above mentioned sentence: [https://doi.org/10.1111/jerd.12837 ]
Our response: Thank you for the suggestion, we added both this reference and rephrased statement. “Nevertheless, another limitation of this study is the lack of cyclic fatigue that could outline even more the behavior of the adhesive cementation interface and the external gap progression [36]." (Line 399-401)
Reference added:
- Baldi, A.; Comba, A.; Ferrero, G.; Italia, E.; Michelotto Tempesta, R.; Paolone, G.; Mazzoni, A.; Breschi, L.; Scotti, N. External gap progression after cyclic fatigue of adhesive overlays and crowns made with high translucency zirconia or lithium silicate. Journal of esthetic and restorative dentistry : official publication of the American Academy of Esthetic Dentistry ... [et al.] 2022, 34 (3), 557-564. DOI: 10.1111/jerd.12837From EBSCOhost MEDLINE Complete.
Reviewer comment #7:
The authors have not treated the specimen with IDS (Immediate dentin sealing) which is the gold-standard for indirect adhesive cementation. This could also have avoided the penetration inside the tubules. Please support the decision not to use IDS.
Our response: Thank you for the comment and addition. It is recommended to use IDS to limit the penetration of temporary cement into the dentinal tubules. However, using dentin sealing would not allow us to answer the question of our study – if temporary cement particles can penetrate the dentinal tubules or not. Another reason for not using IDS was to not compromise the recommendations of the instructions for use of the adhesive cement. It would certainly be interesting to extend the study by analyzing the effect of IDS and the difference in results.
We revised accordingly to your suggestion and have added the following in the discussion part:
“To limit the penetration of temporary cement into the dentinal tubules immediate dentin sealing can be used. However, using dentine sealing would not allow us to answer the question of our study. Therefore, we evaluated how the use of temporary materials, surface pretreatment and the size of the total tooth area (enamel and dentin) available for cementation may impact the bonding between the tooth and the ceramic restoration.” (Lines 282-287).

Reviewer 2 Report
I just have a minor question: how did you establish the number necessary in each group, i.e. 20 for testing and 6 for SEM?
Author Response
Our responses to Reviewer 2
We appreciate your helpful comment which allowed us to improve the manuscript. We used Word track changes so that you can see the manuscript revisions.
Reviewer comment: I just have a minor question: how did you establish the number necessary in each group, i.e. 20 for testing and 6 for SEM?
Our response: Thank you for the question. We chose the number of specimens based on information provided in previous studies describing their experimental setting, including the tensile bond strength test and surface analysis.
We have added the following in the discussion part:
A limitation of our study is the small sample size, this was due to our limited availability of biological material. However, we based the number of specimens needed based on information provided about the settings in previous simple experimental study where the tensile bond strength test and surface analysis were used. (Lines 365-367)

Round 2
Reviewer 1 Report
The authors have amended to all the requests
Author Response
Thank you very much for your review!
Reviewer 2 Report
The questions have been answered
Author Response
Thank you very much for your review!